# Clinical, Radiological, and Echocardiographic Findings in Cats Infected by *Aelurostrongylus abstrusus*

**DOI:** 10.3390/pathogens12020273

**Published:** 2023-02-07

**Authors:** Ettore Napoli, Michela Pugliese, Angelo Basile, Annamaria Passantino, Emanuele Brianti

**Affiliations:** 1Department of Veterinary Sciences, University of Messina—Via Umberto Palatucci, 98168 Messina, Italy; 2Centro Nefrologico Veterinario, 95123 Catania, Italy

**Keywords:** cat, *Aelurostrongylus abstrusus*, cat lungworm, pulmonary hypertension, clinical signs

## Abstract

Cats infected by *Aelurostrongylus abstrusus* may show a plethora of clinical signs, and pulmonary hypertension (PH) seems to be one of the possible alterations induced by the infection; however, data on this association are scant and contradictory. Therefore, the aims of this study were to investigate the association between aelurostrongylosis and PH and to evaluate the correlation between the number of *A. abstrusus* larvae expelled in the faeces and the clinical, echocardiographic, and radiological alterations. Fifteen cats (i.e., eight males and seven females) older than 3 months and naturally infected by *A. abstrusus* with different parasitic loads, expressed as larvae per grams of faeces (l.p.g.), were enrolled in the study. Each animal underwent clinical, echocardiographic, and radiographic examinations. Most cats (i.e., 10/15) showed pathological patterns on thoracic radiograms; particularly, the alveolar pattern (four cats), interstitial-nodular pattern (five cats), and bronchial pattern (one cat). No significant echocardiographic findings of PH were detected. No correlation between the number of l.p.g. and the severity of clinical signs was observed, but a significant correlation with activated partial thromboplastin time (aPTT), radiographic alterations (interstitial nodular pattern), and ultrasonographic findings (RIVIDs) were noticed. These findings suggest that other factors such as animal age and health status, as well as comorbidity, may influence the presentation of the disease or the clinical manifestation and severity of the disease.

## 1. Introduction

Nowadays, nematodes are regarded as the most frequent etiological parasitic agents of gastrointestinal and respiratory diseases in cats [1]. Different Metastrongyloidea species (lungworms) could localize in the respiratory system of cats (i.e., *Aelurostrongylus abstrusus, Troglostrongylus brevior*, and *Oslerus rostratus*) [2,3]. A multicentre survey conducted in Europe demonstrated that feline lungworms are widely diffused, at an estimated prevalence of 20.8% (ranging from 0.8% to 35.8%), with *A. abstrusus* being the most prevalent species [3]. *Aelurostrongylus abstrusus* has an indirect life cycle; indeed, adult stages reside in terminal bronchi, alveolar ducts, and pulmonary alveoli [4]. Though most of the infections are asymptomatic [5], cats infected with *A. abstrusus* could show a plethora of clinical findings characterized by dyspnoea, with abdominal breathing, mucopurulent nasal discharge, and tachypnoea, up to respiratory failure [6,7,8,9] and a fatal outcome [10]. In addition, the clinical presentation of aelurostrongylosis could be complicated by the presence of comorbidity such as FeLV and FIV infections [11]. Among the pathological alterations related to lungworm infection, some authors reported the presence of pulmonary hypertension (PH) [9,12,13]. PH is a pathological condition common in dogs [14] but very rare in cats being observed in the course of heartworm infection, thromboembolic disorders, inflammatory polyps, and patent ductus arteriosus [15,16,17,18]. Although in some cases PH has been reported in the course of *A. abstrusus* infection, the association between PH and this lungworm species is still debated [9,12,13]. Indeed, in a recent clinical study that investigated the association between *A. abstrusus* natural infection and PH, none of the fourteen enrolled patients showed PH; however, data on the parasite load of infected cats were not reported [9]. Nevertheless, as stated elsewhere, it seems that the clinical presentation of aelurostrongylosis could be influenced by the parasite intensity [10], and it is still unclear if a correlation between the number of first-stage larvae (L1) expelled with the faeces and the severity of clinical signs exists [11,19]. Similarly, it is yet not certain if the number of larvae expelled with the faeces could be used as a parameter for the prediction of lung alterations detectable using thoracic radiograms. Based on the above observations, the aims of the study were (i) to explore the association between *A. abstrusus* infection and PH in naturally infected cats and (ii) to investigate the correlation between the number of *A. abstrusus* larvae excreted with the faeces and echocardiographic and radiological alterations. 

## 2. Materials and Methods

### 2.1. Ethical Statement

The protocol and study design were approved by the Ethical Committee of the Department of Veterinary Sciences, University of Messina (Italy) (Approval No. 19; date of approval 10 January 2018). All cat owners were informed of the study aims and the clinical procedures and provided a signed informed consent before enrolment.

### 2.2. Study Population and Laboratory Procedures 

A total of 70 cats routinely referred to the Veterinary Teaching Hospital of the University of Messina, Italy, were tested for lungworm presence using the Baermann–Wetzel method [20]. Larvae retrieved with this method were identified at the species level using morphological keys [21], and their load was expressed as larvae per gram of faeces (l.p.g.).

Fifteen (21.4%) of the tested cats scored positive for *A. abstrusus* presence and were enrolled in the study. Each selected cat underwent a physical examination and radiographic and echocardiographic evaluation. Additionally, in the week before the clinical and ultrasonographic examinations, three faecal samples were collected again and analysed using the Baermann–Wetzel method to confirm the presence of lungworms and to assess the parasitic load expressed as larvae per grams of faeces (l.p.g.).

A blood sample of 5 mL was collected from the jugular vein and divided into K_3_EDTA, sodium citrate 3.2%, and dry tubes depending on the parameter to be analysed. 

A complete count of blood cells, including red blood cells (RBC), packed-cell volume (PCV), and white blood cells (WBC) with differential white blood cell count and platelets (PLT) was performed using an automated haematology analyser (ProCyte Dx, IDEXX Laboratories, Westbrook, Maine, USA). In addition, haemoglobin (Hgb) was evaluated. Samples collected in sodium citrate tubes were immediately analysed to determine the activated prothrombin time (aPT) and activated partial thromboplastin time (aPTT) (Coag Dx, IDEXX Laboratories, Westbrook, ME, USA). 

The serum obtained with centrifugation was tested for feline immunodeficiency virus (FIV) and feline leukaemia virus (FeLV) using rapid ELISA assays (IDEXX SNAP® FIV/FeLV Combo Plus Test).

### 2.3. Clinical and Imaging Score

The clinical presentation of *A. abstrusus* infection was assessed using a numeric scoring of symptoms (i.e., Cough, Increased vesicular breath sound, Crackles, Wheezing, Dyspnoea, Sneezing, Nasal discharge, Lymph enlargement). Briefly, each sign was assessed independently of the others and classified as 0 absent, 1 slightly manifested, 2 moderately present, and 3 severely shown. Thoracic radiographs in latero-lateral and dorso-ventral views were carried out. X-ray examinations were carried out blindly and separately by two different veterinarians highly trained in imaging to assess possible radiological alterations. Radiological alterations were scored for bronchial, alveolar, and interstitial (i.e., nodular and reticular) patterns [10,22]. The bronchial pattern was ranked as 0 no changes, 1 mild changes (i.e., first-generation bronchi visible), 2 medium changes (i.e., second-generation bronchi visible), and 3 severe changes (i.e., third-generation bronchi visible). The alveolar pattern was ranked as 0 no changes, 1 mild changes (i.e., isolated fluffy infiltrates), 2 medium changes (i.e., well-defined with air bronchograms), and 3 severe changes (i.e., lobar signs). Reticular and nodular interstitial patterns were classified as 0 no changes, 1 mild changes (i.e., interstitial framework visible but not distinguishable from the bronchial pattern), 2 medium changes (i.e., interstitial framework visible and distinguishable from the bronchial pattern), and 3 severe changes (i.e., reticular interstitial pattern). The vascular pattern was also evaluated as enlargement of pulmonary arteries and/or veins as 0 no changes, 1 mild changes, 2 medium changes, and 3 severe changes.

### 2.4. Echocardiography

All the cats underwent transthoracic echocardiographic examination using an ultrasound machine (Vivid i Q Premium, General Electric Company, Boston, MA, USA) equipped with phased-array transducers (5–12 MHz) and a simultaneous single-lead electrocardiogram. 

To avoid bias related to animal movement, all the patients were sedated with butorphanol (i.e., 0.2 mg/kg, IM) before echocardiography [23].

Cats were imaged from the right and left parasternal positions, and standard echocardiographic two-dimensional, M-mode, and Doppler images were acquired. All echocardiographic assessments, measurements, and calculations were performed using a digital offline workstation (EchoPAC, General Electric Company, Boston, MA, USA). Values for each echocardiographic variable were calculated as the average of three measurements.

Whenever tricuspid regurgitation (TR) or pulmonary regurgitation (PR) were present, peak velocities were measured. PH was defined as a peak TR velocity of >2.7 m/s and/or a peak PR velocity of 2.2 m/s [24]. Systolic time intervals of pulmonary flow (i.e., acceleration time (AT), ejection time (ET), and AT/ ET ratio) were acquired from the right parasternal short-axis view at the level of the heart base using pulse-wave Doppler flow signals and small sample volume size (2–3 mm). The pulmonary artery flow (PFP) profile was defined as type I (i.e., normal flow profile, symmetric envelope, and similar acceleration and deceleration times), type II (i.e., abnormal flow profile, asymmetric envelope, and peak velocity occurring early in systole with a longer deceleration time), and type III (i.e., abnormal flow profile, asymmetric envelope, and rapid acceleration with notching during deceleration) [25]. Maximum right atrial diameter (RA) was measured at end-systole from the mid-point of the interatrial septum to the right atrial lateral wall in a cranial–caudal plane and parallel to the tricuspid valve annulus. The presence of interventricular septal flattening (ISF) was subjectively assessed from standard right-parasternal short-axis views and classified as either present (1) or absent (0) independent of the phase of the cardiac cycle. From the right parasternal short-axis heart base view, measurements of the early diastolic internal dimensions of both the pulmonary artery (PA), at the level of the pulmonary valve, and the aortic valve dimension (Ao), from the blood–tissue to the blood–tissue interface, were obtained. The ratio PA/Ao was calculated.

Right ventricular fractional area change (FAC) and tricuspid annular plane systolic excursion (TAPSE) measurements were acquired from a left apical 4-chamber view optimized for the right ventricle. For the FAC calculation, measurements of the right ventricle area were obtained by tracing the right ventricular endomyocardial border at end-diastole (RVIDd) and end-systole (RVIDs), excluding the papillary musculature. Percent FAC was derived using the following formula: (RVIDd − RVIDs)/RVIDd × 100. TAPSE was measured from an M-mode recording of the lateral aspect of the tricuspid valve annulus. To limit the risk of underestimating the measurement, the M-mode cursor was carefully aligned parallel to the longitudinal displacement of the tricuspid valve plane with the right ventricular free wall. TAPSE was measured with electronic callipers between the most basilar position of the tricuspid annulus at end-diastole and its most apical displacement at end-systole using the leading-edge method. The TAPSE values were averaged from measurements of three consecutive beats during sinus rhythm. 

As reference values, the control groups from the study of Visser and collaborators [26] and Vezzosi and Schober [27] were used.

### 2.5. Statistical Analysis

Numerical variables were expressed as mean and standard deviation (SD), while categorical parameters were classified as levels. The Kolmogorov–Smirnov test for normality was carried out to assess the normal distribution. The Pearson’s test or Spearman’s correlation test, as appropriate, was performed to assess significant correlations between l.p.g. and clinical and radiological scores; moreover, the same test was used to assess the correlation between l.p.g. and the changes in the echocardiographic variables. 

Significance was set at *p* < 0.05. All the statistical analyses were performed using GraphPad Prism version 8.00 (GraphPad Software, San Diego, CA, USA).

## 3. Results

All the enrolled *A. abstrusus*-infected cats were European Shorthair, older than 3 months, in good general health condition, and not affected by chronic or debilitating diseases (Table 1). All the cats had regular outdoor access and were co-housed with either co-specific and/or hetero-specific animals (i.e., dogs and rabbits). Moreover, for most of them, predatory activity was observed and reported by the owner. All the cats tested negative for both FeLV and FIV using the ELISA assay. 

The number of *A. abstrusus* larvae expelled with the faeces ranged from 10 to 5900 l.p.g. (Table 1). The majority of patients were clinically asymptomatic (9 out of 15), while 6 cats showed some clinical signs, the commonest being lymphadenomegaly (6 out of 15) followed by dyspnoea (4 out of 15). 

Alterations in the haematological parameters or coagulative profile were observed both in the clinically symptomatic and asymptomatic cats (Table 2). A reduction in the number of basophils (in 8/15 cats) and eosinophil (3/15 cats), except for 1 cat (i.e., GA1) in which a slight increase was observed. In three cats, an increase in the number of monocytes and in the Hgb and PCV values was also observed. Thrombocytopenia was observed in 7 out of the 15 cats. The coagulation parameters were observed in six patients; particularly, in five cats, both aPTT and aPT were altered, while one cat (GA8) showed only a prolongment of aPT (Table 2). 

In the majority of cats (10 out of 15), even if no respiratory signs were present, the thoracic radiograms showed pathological changes. Specifically, an alveolar pattern was observed in four out of the fifteen cats, an interstitial nodular pattern and an interstitial reticular pattern were observed in three and two patients, respectively, while a bronchial pattern was detected in one cat. The most serious radiological alteration was observed in one cat (GA1) with the highest l.p.g. count (i.e., 5900 ± 26 l.p.g.); particularly, in this cat, an interstitial nodular pattern was observed (Figure 1) that was classified with the 3rd grade of severity.

Data regarding echocardiographic parameters are summarized in Table 3. All parameters were within the normal ranges for the species regardless of the number of L1 expelled with the faeces except one cat (GA6) that showed TR but with a normal value of PR.

PH was not detected in any of the studied cats, nor was there a correlation between the l.p.g. and the presence and severity of clinical signs, or an alteration in the haematological and coagulative values (Table 4), with the exception of aPTT. In addition, a significant positive correlation (i.e., r = 0.5422 and *p* = 0.0396) between the presence of interstitial nodular pattern and the number of *A. abstrusus* L1 expelled was observed (Table 4). Regarding the echocardiographic parameters, no correlation with the larval emission rate (Table 4), except for the RVIDs (i.e., r = 0.2655 and *p* = 0.0493), were observed.

## 4. Discussion

The present study evaluated the correlation between the number of *A. abstrusus* larvae expelled with the faeces and clinical, haematological radiological, and echocardiographic changes in naturally infected cats demonstrating that the number of *A. abstrusus* larvae is not correlated with clinical signs nor haematological changes, except for aPTT. On the other hand, the presence and severity of radiological alterations correlated with the number of L1 expelled with the faeces. In addition, none of the infected cats showed PH signs and, therefore, no association between *A. abstrusus* infection and PH was detected. 

This study was conducted in a heterogeneous population composed of animals from both sexes ranging from juvenile to adult patients. The heterogenicity of the study population guarantees that the results drawn are not biased by the selected animals. Moreover, all the studied cats tested negative for FeLV and FIV, two debilitating factors that could impair the animal’s health worsening the clinical manifestation of aelurostrongylosis [10].

However, during the physical examination, the majority of the cats, regardless of the number of L1 expelled with the faeces, scored as clinically asymptomatic, as reported either in experimental infections [28] or in naturally infected animals [10].

According to the obtained results, no correlation between the number of L1 shed with the faeces and the severity of clinical signs and the degree of lung damage exists, with the exception of some alterations observed radiologically (i.e., the presence of an interstitial nodular pattern). Other factors, such as age, sex, lifestyle, and the concomitant presence of other infective and debilitating diseases, such as FeLV and FIV, have to be taken into account to understand and predict the severity of the clinical presentation [3,10,11,13,19,28,29,30,31,32].

In the majority of symptomatic cats, vague, non-respiratory symptoms, i.e., lymphadenopathy, were observed; conversely, specific signs were detected in a few patients, with dyspnoea being the most frequent. In addition, in agreement with a previous study [10], other respiratory signs such as increased vesicular breath sound, crackles, wheezing and sneezing were less frequently detected. Interestingly, the majority of symptomatic patients shed a low or medium number of L1, corroborating, once again, the finding that clinical signs are not correlated with the number of expelled larvae. In fact, the number of larvae shed with the faeces is not always indicative of the number of adult worms or L1 present in the terminal bronchioles and alveoli, especially if the intermittence in larvae shedding is taken into account [33]. 

Regarding the haematological alterations, non-specific haematological alterations have been described in the course of aelustrongylosis so far. Polycythemia, the increase in PCV, determines thickened blood, and slow flow, and may cause an increase in clot-forming risk; it may occur in response to some known stimulus such as hypoxia [34]. 

Eosinophilia has been reported as the most frequent finding occurring in cats naturally or experimentally infected by *A. abstrusus*, presumably related to the continuous antigen stimulation produced by the presence of adult nematodes and larvae [6,8,28], while monocytosis and basophilia have been occasionally reported, likely because of chronic inflammation [11]. 

In the studied cats, we did not observe the presence of haemorrhagic diathesis, but the prolongment of aPT and aPTT observed in the cats examined suggest an increased consumption of coagulation factors, associated in any case with platelets consumption indicated by the onset of thrombocytopenia, as reported elsewhere [35]. This finding suggests the presence of a silent or subclinical disseminated intravascular coagulation as a consequence of the arterial endothelial damage caused the *A. abstrusus* release of endogenous vasoactive compounds [36].

In fact, in some cats that did not show alterations in aPT and aPTT, thrombocytopenia was observed. This last finding would support the hypothesis of the onset of immune-mediated thrombocytopenia similar to that observed in dogs infected with *Angyostrongylus vasorum* [37]. 

Interestingly, the absence of clinical signs and the low relevance of the haematological alterations did not correspond to the radiographic findings. In fact, in asymptomatic patients, radiological alterations of different degrees of severity were observed, thus corroborating the previous observations for both *A. abstrusus* and *T. brevior* [10,11,19,38]. 

In general, as reported elsewhere, the radiological presentation of aelurostrongylosis is non-specific and could be indicative of other lung disorders such as pulmonary fibrosis, pulmonary edema, fungal disease, asthma, and neoplasia [19,39]. However, as corroborated by the results of this study, radiological alterations are those that better reflect the severity of the infection. Specifically, an alveolar pattern prevails in the heaviest larval shedding (5–15 weeks post-infection) followed by bronchial and interstitial patterns in the chronic phase of infection [10]. The results herein presented are consistent with other studies and indicate that interstitial and bronchial patterns are the commonest radiological alterations in aelurostrongylosis [6,11,19]. Radiographic changes, such as the nodular pattern in the lung, may appear before clinical signs and may be correlated with the number of L1 expelled with the faeces [10]. Therefore, thoracic radiography could not only generate a clinical suspicion [10,38] of lungworm infection but also provide valuable information on the severity of the infection. 

In the cats studied in the present study, PH was not observed. According to Lacava et al. 2017 [19], cats naturally infected with *A. abstrusus* show a variety of clinical and radiographic findings but not PH. As it is commonly accepted, PH is a very rare pathological condition in cats, generally associated with heartworm infection, thromboembolic disorders, inflammatory polyps, and *patent ductus arteriosus* [15,16,17,18]. PH is more common in dogs, which may be associated with the presence of the metastrongyloid nematode *A. vasorum* [40,41,42]. Concretely, 14.6% of dogs infected with this parasite develop moderate-to-severe PH [41]. In cats, PH was recently associated with lungworm infection caused by *A. abstrusus* [9,13] and *T. brevior* [12]. However, in the case of *A. abstrusus*, PH was observed only in few cases, i.e., a 6-month-old kitten [13] and two patients of 2 months [9]. In the same manner, the cat affected by *T. brevior* that showed irreversible pulmonary hypertension was a young patient of 4 months [12]. In the current study, however, PH was not observed either in those cats shedding a low number of larvae in the faeces nor in those featured by a very high l.p.g. count.

Based on the literature and on the results herein presented, it is not possible to associate PH with *A. abstrusus* regardless of the number of L1 shed in the faeces.

It seems that PH occurring in the course of feline aelurostrongylosis is related to an increase in pulmonary vascular resistance following the hyperplasia of smooth muscle cells and hypertrophy of the media and intima [36]. The rare presence of PH, observed only in kittens or young patients, suggests that the possible onset is likely related to the small dimension of the pulmonary vessel whose hemodynamics would be more subject to the modifications linked to the reductions in diameter.

Therefore, the results of the present study suggest that PH is not a common alteration in feline aelurostrongylosis, while, by contrast, a radiographic investigation may demonstrate severe pulmonary alteration even in clinically asymptomatic patients.

## 5. Conclusions

In conclusion, for the management of aelurostrongylosis, it is important to use an integrated approach that combines sensitive copromicroscopy methods (i.e., the Baermann–Wetzel technique), with blood chemistry and radiological investigations. Only after having carried out a correct diagnostic procedure, it is possible to formulate a reliable prognosis and draw up the most suitable treatment plan for the patient. It is mandatory for the veterinary practitioner not only to include *A. abstrusus* differential diagnosis in cats that shows clinical signs but to systematically test all the animals potentially exposed to the parasite and particularly those more susceptible such as, for instance, young animals with regular outdoor access.

## Figures and Tables

**Figure 1 pathogens-12-00273-f001:**
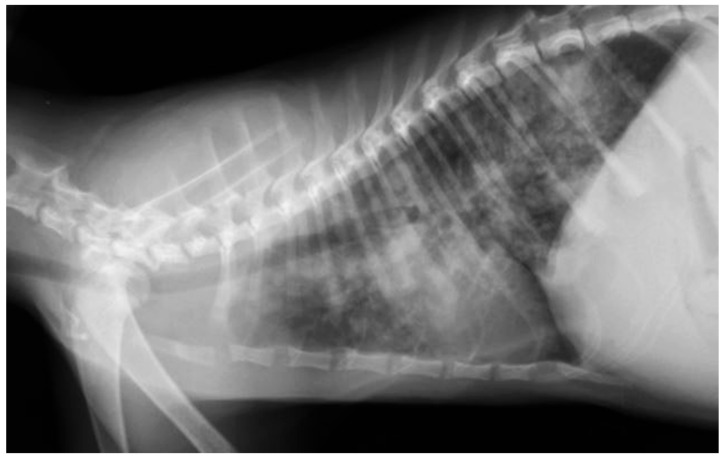
Latero-lateral X-ray in a cat (GA1) naturally infected with *Aelurostrongylus abstrusus* showing an interstitial nodular pattern.

**Table 1 pathogens-12-00273-t001:** Signalment and clinical scoring in examined cats that were naturally infected with *Aelurostrongylus abstrusus*.

ID	L.p.g.	Age	Sex	BCS	PR	RR	Cough	VBS	Crackles	Wheezing	Dyspnoea	Sneezing	Nasal Discharge	Lymphadenomegaly	Total Score
GA1	5.900 ± 26	4	M	4	110	40	2	2	2	1	2	1	0	2 ^R^	12
GA2	26 ± 4	13	M	5	132	20	1	1	0	1	1	1	2	2 ^A^	9
GA3	40 ± 2	36	F	5	148	23	1	2	2	1	2	2	2	2 ^A^	13
GA4	80 ± 12	18	M	5	147	22	0	0	0	0	1	0	0	2 ^A^	3
GA5	60 ± 22	56	M	5	110	24	0	0	0	0	0	0	0	0	0
GA6	600 ± 47	72	M	4	140	20	0	1	0	0	0	0	0	1 ^A^	2
GA7	100 ± 18	12	F	5	100	52	0	0	0	0	0	0	0	0	0
GA8	300 ± 26	8	M	4	120	20	0	0	0	0	0	0	0	0	0
GA9	10 ± 7	24	M	6	120	30	0	0	0	0	0	0	0	0	0
GA10	460 ± 41	36	F	5	135	32	0	0	0	0	0	0	0	2 ^R^	2
GA11	26 ± 11	36	F	4	138	34	0	0	0	0	0	0	0	0	0
GA12	189 ± 9	5	F	4	140	30	0	0	0	0	0	0	0	0	0
GA13	40 ± 4	12	M	4	128	35	0	0	0	0	0	0	0	0	0
GA14	2000 ± 18	13	F	5	120	38	0	0	0	0	0	0	0	0	0
GA15	1500 ± 136	18	F	6	128	36	0	0	0	0	0	0	0	0	0
Mean	755.4	24.2	-	4.7	127.7	30.4									
SD	1440.97	18.3	-	0.66	13.32	8.7									
Max	5900	72	-	6	148	52									
min	10	4	-	4	100	20									

Lpg = Larvae per gram of faeces; Age (months); BCS = Body Condition Score; PR = Pulse Rate (beat per minutes); RR = Respiratory Rate (breaths per minute); VBS = Vesicular Breath Sounds; M = Male; F = Female; SD = Standard Deviation; ^R^ Retropharingeal lymph nodes; ^A^ Axillary lymph nodes.

**Table 2 pathogens-12-00273-t002:** Haematological and coagulative variables in cats infected with *Aelurostrongylus abstrusus*.

ID	RBCs	PCV	Hgb	WBCs	NEU	LYMPH	MON	EOS	BAS	PLT	aPTT	aPT
NV	(6.64–12.20 M/mmL)	(30.3–52.3%)	(9.8–16.2 g/dL)	(2.87–17.02 K/mmL)	(1.48–10.29 K/mmL)	(0.92–6.88 K/mmL)	(0.05–0.67 K/mmL)	(0.17–1.57 K/mmL)	(0.01–0.26 K/mmL)	(151–600 K/mmL)	(65–119 s)	(15–22 s)
GA1	7.5	31.8	10.1	15.52	10.29	2.43	0.67	1.65 ^(>5.1%)^	0.007 ^(<30%)^	366	300 ^(>152.1%)^	29 ^(>31.8%)^
GA2	7.57	38.6	10.1	9.13	6	2.08	0.26	0.58	0 ^(<100%)^	112 ^(<25.8%)^	130 ^(>9.2%)^	26 ^(>18.2%)^
GA3	6.64	36	11.9	8.04	5.6	1.62	0.71 ^(>5.9%)^	0.01 ^(<100%)^	0.1	47 ^(<68.8%)^	122 ^(>2.5%)^	28 ^(>27.3%)^
GA4	7.74	48.7	18.7 ^(>15.4%)^	8.48	5.94	1.82	0.68 ^(>1.5%)^	0 ^(<100%)^	0.04	209	115	19
GA5	7.05	47.8	16.7 ^(>3.9%)^	10.4	7.2	0.92	1.58 ^(>135%)^	0.7	0.02	263	98	16
GA6	6.70	41.0	14.1	10.49	6.38	1.6	0.4	1.3	0 ^(<100%)^	178	68	22
GA7	7.57	38.6	10.1	9.13	6	2.08	0.26	0.58	0 ^(<100%)^	112 ^(<25.8%)^	130 ^(>9.2%)^	26 ^(>18.2%)^
GA8	7.76	57.9 ^(>10.7%)^	19 ^(>17.3%)^	9.64	5.16	2.65	0.7	0.2	0 ^(<100%)^	733 ^(>22.1%)^	84	23 ^(>4.5%)^
GA9	9.56	50.3	15.2	5.43	2.91	2.1	0.26	0.12 ^(<29.4%)^	0.04	87 ^(<42.4%)^	88	21
GA10	8.4	40	14.18	9.43	6.06	1.95	0.66	0.48	0.04	236	133 ^(>11.8%)^	23 ^(>4.5%)^
GA11	7.8	32	12	12.3	8.2	0.98	0.56	1.53	0 ^(<100%)^	112 ^(<25.8%)^	78	18
GA12	7.2	36	16	14.5	5.2	3	0.54	0.98	0 ^(<100%)^	156	98	20
GA13	8.9	42	14.2	16,1	9.9	2.9	0.61	0.24	0.03	234	67	16
GA14	6.9	45	13.6	8.3	4.5	1.3	0.14	1.2	0.02	346	85	21
GA15	8.5	38	15.8	7.9	5.7	2.4	0.25	1.1	0 ^(<100%)^	89 ^(<41%)^	110	22
Mean	7.66	41.84	13.99	10.49	6.38	1.96	0.57	0.68	0.02	227.93	114.00	22.00
SD	0.77	6.89	2.76	2.85	1.85	0.61	0.33	0.52	0.03	161.32	54.21	3.85
Min.	9.56	57.9	19	16.1	10.29	3	1.58	1.64	0.1	733	300	29
Max	6.64	31.8	10.1	5.53	2.91	0.9	0.14	0	0	47	67	16
Out of range	0	1	3	0	0	0	3	4	7	8	5	6

NV = Normal Value; SD = Standard Deviation; RBCs = Red Blood Cells; PCV = Packed Cell Volume; Hgb = Hemoglobin; WBCs = White Blood Cells; NEU = Neutrophils; LYMPH=Lymphocytes; MON = Monocytes; EOS = Eosinophils; BAS = Basophils; PLT = Platelet Count; aPTT = Activated Partial Thromboplastin Time; aPT = Activated Prothrombin Time. In brackets is the percentage of departure from the reference values both as a decrease (<) and as an increase (>).

**Table 3 pathogens-12-00273-t003:** Number of larvae expelled with the faeces and echocardiographic variables measured in cats naturally infected with *Aelurostrongylus abstrusus*.

ID	Lpg	TR	PR	PFP	AT	ET	AT/ET	MPAD	RA	RVIDd	RVIDs	FAC	TAPSE	ISF	PA	Ao	PA/Ao
GA1	5.900 ± 26	0	0	1	52	155	0.34	6.38	11.21	0.7	0.3	57.14	10.18	0	4.47	6.51	0.69
GA2	26 ± 4	0	0	1	55	166	0.33	7.8	n.a.	0.8	0.4	50.00	11	0	5.33	6.04	0.88
GA3	40 ± 2	0	0	1	44	137	0.32	n.a.	n.a.	0.8	0.4	50.00	9.9	0	n.a.	n.a.	
GA4	80 ± 12	0	0	1	45	163	0.28	4.1	8.28	0.9	0.5	44.44	10	0	6.1	5.78	1.06
GA5	60 ± 22	0	0	1	49	112	0.44	6.1	9.4	9.8	0.4	95.92	11	0	5.1	6.56	0.78
GA6	600 ± 47	1	0	1	59	104	0.57	9.96	11.22	0.9	0.4	55.56	11.27	0	4.9	5.9	0.83
GA7	100 ± 18	0	0	1	76	194	0.39	5.84	9.55	0.9	0.4	55.56	9.1	0	5.02	6.41	0.78
GA8	300 ± 26	0	0	1	91	193	0.47	7.59	12.58	0.8	0.4	50.00	10.1	0	5.3	5.9	0.90
GA9	10 ± 7	0	0	1	49	210	0.23	n.a.	14.07	0.9	0.5	44.44	11	0	8.07	8.06	1.00
GA10	460 ± 41	0	0	1	43	163	0.26	4.95	11	0.7	0.4	42.86	9.8	0	5.47	7.1	0.77
GA11	26 ± 11	0	0	1	65	195	0.33	5.65	12.25	0.8	0.3	62.50	9.71	0	5.3	5.9	0.90
GA12	189 ± 9	0	0	1	45	195	0.23	6	12	0.8	0.4	50.00	9.5	0	7.2	8	0.90
GA13	40 ± 4	0	0	1	58	319	0.18	5.65	12.25	0.9	0.4	55.56	9.71	0	5.4	6	0.90
GA14	2000 ± 18	0	0	1	49	219	0.22	5.7	11	0.9	0.4	55.56	9.3	0	6.3	5.9	1.07
GA15	1520 ± 136	0	0	1	62	196	0.32	6.3	10	0.8	0.3	62.50	9.12	0	5.2	6.4	0.81

Lpg = Larvae per gram of faeces; TR=Tricuspid Regurgitation (Velocity: 0 ≤ 2.7 m/s; 1 ≥ 2.7 m/s); PR = Pulmonary Regurgitation (Velocity: 0 ≤ 2.2 m/s; 1 ≥ 2.2 m/s); PFP = Pulmonary Artery Flow (Type I = 1; Type II = 2; Type III = 3); AT = Right Ventricular Acceleration Time; ET = Right Ventricular Ejection Time; AT/ET = Ratio between AT and ET; MPAD = Main Pulmonary Artery Diameter; RA = Maximum Right Atrial Diameter; RVIDd = Right Ventricular Internal Dimension in Diastole; RVIDs =Right Ventricular Internal Dimension in Systole; FAC = Fractional Area Change; TAPSE = Tricuspid Annular Plane Systolic Excursion; ISF = Interventricular Septal Flattening; PA = Pulmonary Artery; Ao = Aortic Valve Diameter; PA /Ao = PA and Ao ratio.

**Table 4 pathogens-12-00273-t004:** Correlation and significance between haematological, coagulative, radiological, and echocardiographic variables in cats naturally infected with *Aelurostrongylus abstrusus* with different parasitic loads expressed as larvae per gram of faeces (l.p.g.).

	RBCs	PCV	Hgb	WBCs	NEU	LYMPH	MON	EOS	BAS	PLT	aPTT	aPT
r	−0.1214	−0.3431	−0.3267	0.3535	0.4370	0.1408	−0.07274	0.5990	−0.1932	0.2798	0.8290	0.4594
95% c. i.	−0.597–0.417	−0.727–0.205	−0.719– 0.223	−0.194–0.733	−0.097–0.776	-0.400–0.609	−0.564–0.456	0.125–0.850	−0.641–0.354	−0.271–0.692	0.550–0.941	−0.069–0.787
R squared	0.01474	0.1177	0.1068	0.1250	0.1910	0.01983	0.005292	0.3587	0.03731	0.07827	0.6872	0.2111
*p* value	0.6665	0.2106	0.2346	0.1962	0.1034	0.6167	0.7967	0.1833	0.4903	0.3125	0.0001 *	0.0849
	Bronchial pattern		Alveolar pattern		Interstitial nodular pattern		Interstitial reticular pattern					
r	0.1859		0.05081		0.5422		−0.2101					
95% confidence interval	−0.375–0.647		−0.487–0.560		0.0247–0.830		−0.662–0.353					
*p* Value	0.6667		0.8559		0.0396 *		0.4571					
	AT	ET	AT/ET	MPAD	RA	RVIDd	RVIDs	FAC	TAPSE	PA	Ao	PA/Ao
Pearson r												
r	−0.08826	−0.09779	−0.004329	0.02921	−0.04605	−0.1385	−0.5152	0.04951	−0.1043	−0.3408	−0.05966	−0.3835
95% c. i.	−0.574–0.444	−0.581–0.436	−0.515–0.509	−0.530–0.571	−0.582–0.518	−0.607–0.402	−0.812–0.004	−0.475–0.548	−0.585–0.431	−0.738–0.232	−0.572–0.486	−0.759–0.185
R squared	0.007790	0.009562	1.874005	0.0008534	0.002121	0.01919	0.2655	0.002452	0.01087	0.1162	0.003560	0.1470
*p* Value	0.7544	0.7288	0.9878	0.9245	0.8812	0.6225	0.0493 *	0.8609	0.7116	0.2331	0.8394	0.1759

* indicates the significant correlations; RBCs = Red Blood Cells; PCV = Packed Cell Volume; Hgb = Hemoglobin; WBCs = White Blood Cells; NEU = Neutrophils; Lymph = Lymphocytes; Mon = Monocytes; EOS = Eosinophils; BAS = Basophils; PLT = Platelet Count; aPTT = Activated Partial Thromboplastin Time; aPT = Activated Prothrombin Time; AT = Right Ventricular Acceleration Time; ET = Right Ventricular Ejection Time; AT/E T = Ratio between AT and ET; MPAD = Main Pulmonary Artery Diameter; RA = Maximum Right Atrial Diameter; RVIDd = Right Ventricular Internal Dimension in Diastole; RVIDs = Right Ventricular Internal Dimension in Systole; FAC = Fractional Area Change; TAPSE = Tricuspid Annular Plane Systolic Excursion; PA = Pulmonary Artery; Ao = Aortic Valve Diameter; PA/Ao = PA and Ao ratio.

## Data Availability

Data may be provided after contacting the corresponding author.

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
