# Peer review of "Clinical, Radiological, and Echocardiographic Findings in Cats Infected by Aelurostrongylus abstrusus"

_pathogens, 2023, doi:10.3390/pathogens12020273_

Round 1
Reviewer 1 Report
Dear Authors, I reviewed with pleasure the manuscript entitled: Clinical, radiological, and echocardiographic findings in cats infected by Aelurostrongylus abstrusus.
The manuscript is overall of high quality, it explains the clinical implications of Aelurostrongylus abstrusus in pulmonary hypertension. Even though this parasite is highly studied the aspects related to the PH were not clearly elucidated in the past. In my opinion, the manuscript adds a pleasing "brick" to the state of the clinical findings about this important nematode.
The methodology seems adequate and the clinical and paraclinical tests adopted are satisfactory to sustain the conclusions.
The discussion paragraph is appropriate even if that can be cut off at some point.
English is acceptable and eventually, grammar checks could be done during the proofreading process.
Author Response
Please, see the attachment.

Reviewer 2 Report
I think it is a very interesting study that merits publication. This study provides valuable information on a disease that is poorly recognized in cats, despite the health alterations it can cause in infected cats. The methodology used is good, and the results obtained are well discussed. I only have these considerations to make:
Throughout the text, the authors indicate that the larvae are “emitted” with the faeces. I think it is more appropriate to refer to the larvae being “expelled” or “shed” with the feces.
Line 2: …radiological and…
Line 17: …5.900 larvae per gram of faeces…
Line 18: It is unclear why 13 cats are included in this result when the total number of animals in the study was 15.
Line 19: …(4 cats), interstitial-nodular pattern (3 cats), and bronchial pattern (one cat).
Line 20: No significant echocardiographic findings of PH were detected.
Lines 21-22: Perhaps the authors should be less categorical about this statement. I believe that the factors they indicate may influence the presentation of obvious clinical cases, but we cannot say that they are crucial factors. More research should be done. You also don’t understand what they mean when they say it influences the determinism of the disease. I think it would be more precise to indicate that they could influence the presentation of the disease, or the clinical manifestation and severity of the disease.
Line 45: is still debated
Line 47: …none of the fourteen enrolled patients…
Line 47: …the parasite load…
Line 49: …be influenced by the parasite intensity
Line 51: …severity of clinical signs exists
Line 56: …larvae excreted with the faeces and…
Line 56-57: …echocardiographic and radiological alterations.
Line 66: …Messina (Italy)
Line 67: …retrieved by this method were…
Line 70: Each selected cat underwent physical examination
Lines 72-73: …examinations, three faecal samples were collected and analyses by Baermann-Wetzel method to confirm the presence of lungworms and to assess the parasite intensity.
Line 90: …each sign was…
Line 94: It would be advisable for the authors to indicate whether the operators were veterinarians or specialists in radiological interpretation.
Lines 153-154: …standard deviation (SD), while categorical parameters were classified…
Lines 163, 170, 174, 187, 202 and 218: A. abstrusus in italics.
Line 166: please indicate what animal species other than cat are mainly involved (dog, ruminants, birds…).
Line 171: Lpg= Larvae per gram of faeces;
Line 176: … some clinical signs, being the commonest…
Lines 176-177: I think it is not necessary to repeat “i.e.” when a result is presented enclosed in parentheses, because the interpretation is clear to the reader. This is a comment that can be extended to the rest of the text.
Lines 181-182: … an increase in the number of monocytes and in Hgb and PCV values was also observed.
Line 183: … coagulation parameters were found in six…
Line 187: … variables in infected cats with…
Line 187: Explain in the table footnote what is the meaning of the percentage in parentheses in some of the values found.
Line 191: … respiratory signs were…
Line 192: …changes. Specifically, an alveolar pattern…
Line 194: … bronchial pattern was detected…
Line 198: … in Table 3.
Lines 198-199: … All parameters were within…
Line 202: … measured in naturally infected cats with…
Line 203: the authors should always use the same format for the abbreviation LPG, because in the text and in the tables it sometimes says Lpg or lpg.
Line 210: … clinical signs,
Line 227: … in a cat (GA1) naturally infected with Aelurostrongylus abstrusus showing an interstitial nodular pattern.
Lines 230-235: I recommend that the authors rewrite these sentences, because they repeat the information in some aspects.
Line 242: … ranging from juvenile to adult patients.
Line 243: … are not biased by the selected animals.
Lines 244-245: … two debilitating factors that could enchase the severity of the lesions and, if concomitant with A. abstrusus, could worsen the clinical manifestation of aelurostrongylosis.
Line 253: … the data herein presented suggest that…
Line 260: … specific signs were…
Line 262: … respiratory signs such as increased vesicular breath sound, crackles,…
Line 265: …clinical signs are not…
Lines 265-266: In fact, the number of larvae shed with the faeces is not always indicative of the number of adult worms or L1s present in the terminal…
Lines 265-266: Rewrite the first sentence, because it is redundant.
Line 274: … Eosinophilia has been reported as the…
Line 275: … infected by A. abstrusus, presumably…
Line 276: … presence of adult nematodes and larvae
Line 277: … basophilia have been occasionally…
Line 279: … of the studied cats… (I suggest that the authors use the term "enrolled cats" less frequently throughout the text).
Line 283-284: … coagulation that could be the consequence of arterial endothelial damage caused by A. abstrusus larvae, although the etiopathogenesis of this process deserves to be better investigated. (A comment: I recommend that the authors include some reference to support this hypothesis, either from studies carried out with A. abstrusus or with other cat lungworms where the effect caused by the larvae has been described).
Line 287: …was observed; this last finding
Line 289: …absence of clinical signs and…
Line 291: …patients radiological alterations…
Line 295: …asthma and neoplasia
Line 296: …of this study, radiological…
Line 297: …infection. Specifically, these alterations are…
Lines 300-301: …and indicate that interstitial…
Lines 303-304: …in the lung, may…
Line 308: …[18], cats naturally infected with A. abstrusus…
Line 309: …findings, but not PH.
Line 313: …[40-42]; concretely, 14.6% of dogs infected with this parasite develop…
Line 315: …associated with lungworm infection caused by A. abstrusus…
Lines 315-316: Delete (Crisi et al., 2015).
Line 317: …and two patients of…
Line 322: …mean 24 months)
Line 327: …presented, it is not possible…
Line 332: …patients suggests that the possible onset…
Line 334: …in their diameter.
Line 335: …collaborators [12], PH was diagnosed…
Line 342: …aelurostrongylosis and, by contrast, the results…
Author Response
Please, see the attachment.

Reviewer 3 Report
Dear Authors,
Your manuscript “Clinical, radiological, and echocardiographic findings in cats 2 infected by Aelurostrongylus abstrusus” offers some additional observations on aelurostrongylosis. I would consider it worthy publishing after some revision.
Please see my comments and suggestions below.
L18 Please correct the numbers in accordance with Lines 191-197.
L22. Among the other factors: what about idiosyncratic/genetic background?
L28. Please correct, this should be the superfamily Metastrongyloidea https://www.ncbi.nlm.nih.gov/Taxonomy/Browser/wwwtax.cgi?mode=Tree&id=6310&lvl=3&lin=f&keep=1&srchmode=1&unlock
L50. “of larvae” please change to read “of first stage larvae (L1)” Also, please correct throughout the text L1s to L1, because L1 stands for "first stage larvae" so no further plural should be applied.
L70-71. Please rephrase as physical examination cannot include clinical history.
L78. Please separate Hgb count from blood cells count
L93. Better use consistent terminology: Radiograph or x-ray examination.
In Table 4, the factor "number of emitted L1 -l.p.g." is not mentioned. So, it is not obvious against what parameter the other findings (i.e. hematological, coagulative, radiological, and echocardiographic variables) are correlated. Please edit.
L212. Capital or minuscule p? please be consistent (see M&M).
Table 4. Please indicate (e.g. with an asterisk and its footnote explanation) the statistically significant results.
L230-233. This is a repetition of the aim of the study, stated in the Introduction. Please delete.
L234-240. Please do not repeat the results in the Discussion.
L248. With the exception of the youngest cat (GA1) the conclusion stated here is not clear by the results, and only a statistical analysis could show it. Please, either apply statistics to state this or do not present it as a conclusion of the present study.
L253-255. "Any correlation / exists and..." It is not clear what the Authors intent to say. Please rephrase.
L261. “other”: other than what?
L270. Polycythemia means a high concentration of red blood cells. That is cells (PCV), not the protein Hgb. Please delete.
L276. Please delete “and larvae” or change to “adult nematodes and larvae”.
L284-285. A. abstrusus L1 do not enter the arteries. What could be the mechanism or pathogenesis that links A. abstrusus with endothelial damage? Please discuss and cite related research.
L287-288. “immune-mediated reaction”... resulting to? Please complete your suggestion.
L313. Angiostrongylus vasorum, Please abbreviate.
L324. “:…PH in the patient.” Please add the reference, it is not clear with survey the Authors are referring to, here.
L327. “Therefore”. Do not start a new paragraph.
L334-336. This is not relevant here because in that case PH was reported in troglostrongylosis. It is confounding to mention it at this specific point of the Discussion, because the reader would believe that you are commenting on aelurostrongylosis cases. Please explain the value of TR and PR combination in another way.
L337. “should be further confirmed through PR evaluation” Please add a reference for this statement.
L341-344...."of the present study suggest that PH is not a common alteration in feline aelurostrongylosis while by contrast ... "
L341. "Rule out" is a very strong statement that cannot be supported by the results of a single survey, especially when there are surveys [9,13], discussed and interpreted (on the basis of [43]) here in, that presented different results. It could be rephrased to: “the results of the present study suggest that PH is not a common finding in feline aelurostrongylosis while by contrast, radiographic investigation may demonstrate severe pulmonary alteration even in clinically asymptomatic patients.”
Conclusions: In the opinion of this Reviewer, these Conclusions were not drawn by the results of the present study. For example:
1. nothing has been presented on different treatment plans for aelurostrongylosis. What are those different plans and what are the criteria of selecting what to apply?
2. it is not clear how blood chemistry would help diagnosis
3. it is not clear what a "correct diagnostic procedure" is (given that a positive faecal examination provides diagnosis)
4. the value of regular testing the animals at infection risk and of considering aelurostrongylosis in differential diagnosis are not conclusions of the present study. These would be conclusions of, e.g., an epidemiological study.
Please correct Aelurostrongylus abstrusus to italics (multiple places throughout the text and tables).
English language needs a thorough revision, preferably by a native speaker or someone with similar language skills. Some examples that however do not exhaust the cases are:
· Therefore, the aims (L13)
· Per gram (L17)
· Alteration (L28) Please select another word to express disease.
· Aelurostrongylus abstrusus is featured by an indirect life cycle, whose adults localize in terminal bronchi, alveolar ducts, and pulmonary alveoli (L33-34)-rephrase.
· However instead of despite (L47)
· …between (delete “the”) A. abstrusus (L54)
· independently by the others: replace “by” with “of” (L91)
· lymph adenomegaly should be one word (L176)
· significative should read significant (L211)
· particular to be changed to typical (L228, legend)
· asymptomatic (L251)
· a few (L260)
· “In any of the enrolled cats was observed the presence of hemorrhagic diathesis” incorrect negation (279)
· In any case (singular) (L281)
· “likely showed” change to “indicated”
· Noteworthy is an adjective, so it needs a noun. So, here you should use an adverb such as Interestingly (L289).
· may be appear: delete “be” (L303).
· In any of the cats enrolled in the present study PH was observed. Incorrect negation (L307).
· in any of the cats examined in the study by Lacava and collaborators PH was observed. Incorrect negation (L319).
· Replace “as” with “that” (L332)
· “One cat” instead of “a cat” (L337-338)
Author Response
Please, see the attachment.

Round 2
Reviewer 3 Report
Dear Authors,
Thank you for your kind words regarding this Reviewer's suggestions.
However, unfortunately, I realise that despite the fact that in your letter you replied positively, accepting all of my suggestions, in fact, in the manuscript only some of the recommendations have been incorporated. More strangely, some parts of the manuscript highlighted in red letters, indicating that there have been corrections, are in fact identical to the initial version, misleading about the revision extension.
Of course, the Authors are not obliged to accept all of the Reviewer's suggestions, and they can decline, providing suitable reasoning. Strangely, in the present case, the Authors agreed to all recommendations but applied only some of them. Please see the attached file (the PDF of your point-to-point responses) with the indications of the corrections that have not been addressed (as comments in the PDF) and some additional suggestions.

Round 3
Reviewer 3 Report
Dear Authors,
Thank you for your replies.
I can see that there has been some effort to improve the manuscript. But unfortunately, not enough attention has been paid and a lot of mistakes due to oversight remain. Furthermore, and more important: there is some erroneous reference to the literature, at least to the best of my understanding (please see the last comment).
Table 4: There are more P values that indicate significant correlations (<0.05) than just the one for the Interstitial nodular pattern, indicated with an asterisk by the Authors.
RVAd Is written twice in the footnote of Table 4.
RVAd and RVAs are explained as abbreviations in the footnotes of Tables but are abbreviated differently in the Tables (different letters are used in each of Tables 3 and 4).
aPT= Activated Prothrombin Time is in the footnote of Table 4 but PT is in the Table.
EOS is written in capital letters in the table but not in the footnote.
Symbols < and > are wrong in some spots of Table 2.
Line 243 both sexes and ranging from ranging from juvenile to adult patients
Line 292: aelustrongylosis
Lines 295-296. “…alterations are those that better reflect the severity of the infection. Specifically, these alterations are usually related to the phase and severity of infection,…” why say the same thing 2 times?
Line 307. Reference [18] is not correct here.
Lines 317-318: repetition of the information of lines 306-308.
“A reduction of the number of basophils (in 8/15 cats) and eosinophil (3/15 cats), except one cat (i.e., GA1) in which a slight increase was observed. “ However, in Table 4, P value for EOS is 0.0183, i.e. statistically significant. Which correct? If the P value shows statistical correlation, why isn’t this parameter presented in Results and in the Discussion as statistically significant?
aPT does not show a statistically significant correlation according to the P value shown in Table 4 (0.084). Why is it discussed as such? (“…the number of A. abstrusus larvae is not correlated with clinical signs nor haematological changes, except aPT and aPTT”.)
“Polycythemia, considering an increase of PCV is a consequence of thickened blood”. The opposite is true. Thickened blood is a consequence of polycythemia.
I am not a native English speaker, but I can see that there have been no adequate (if at all) corrections to the language. Editing from a native speaker is obviously required.
Again, the Authors did not correct the negations:
-In any of the studied cats was observed
-the data herein presented suggest that any correlation between the number of L1 shed with the faeces and the severity of clinical signs and the degree of lungs damage exists
-in any of the cats examined in the study by Lacava and collaborators PH was observed
What does enchase mean? (L 246)
Line 293 lung affection. Affection means fondness. It does not make sense.
“…damage caused by A. abstrusus larvae for release of endogenous vasoactive compounds determining ultrastructural changes in pulmonary arteries”. It does not make sense.
I have suggested to add a reference to the sentence “However, TR is an indirect parameter that gives only a suspicion of PH and should be further confirmed through PR evaluation”, because to my opinion this statement is not correct (at least I do not know any publication supporting it) and as such it is not a fair criticism to the paper of Crisi et al. The Authors added reference [27]. However, to the best of my understanding, nowhere in this reference does it say that “PH and should be further confirmed through PR evaluation”, on the contrary, it is only mentioned “…pulmonary hypertension (estimated echocardiographically by a tricuspid regurgitation pressure gradient >36 mmHg)”, while Pulmonary Regurgitation is nowhere mentioned. To my opinion, the whole suggestion that TR is not a suitable parameter for PH evaluation should be omitted, because according to the literature it is.
